# Primary Amenorrhea in Adolescents: Approach to Diagnosis and Management

Laura Gaspari [1,2,3], Françoise Paris [1,2,3], Nicolas Kalfa [2,4,5] and Charles Sultan [1,*]

1    Unité d'Endocrinologie-Gynécologie Pédiatrique, Service de Pédiatrie, CHU Montpellier,
     University of Montpellier, 34295 Montpellier, France
2    Centre de Référence Maladies Rares du Développement Génital, Constitutif Sud, CHU Montpellier,
     University of Montpellier, 34295 Montpellier, France
3    INSERM 1203, Développement Embryonnaire Fertilité Environnement, University of Montpellier,
     34295 Montpellier, France
4    Département de Chirurgie Viscérale et Urologique Pédiatrique, CHU Montpellier, University of Montpellier,
     34295 Montpellier, France
5    UMR 1302 Institute Desbrest of Epidemiology and Public Health, INSERM, University of Montpellier,
     34295 Montpellier, France
*    Correspondence: pr.charles.sultan@gmail.com

**Abstract:** Primary amenorrhea (PA) describes the complete absence of menses by the age of 15 years. It is a devastating diagnosis that can affect the adolescent's view of her femininity, sexuality, fertility and self-image. A normal menstrual cycle can occur only in the presence of: a properly functioning hypothalamus–pituitary axis, well-developed and active ovaries, outflow tract without abnormalities. Any dysfunction in any of these players can result in amenorrhea. PA evaluation includes the patient's medical history, physical examination, pelvic ultrasonography and initial hormone evaluation, limited to the serum-follicle-stimulating hormone (FSH) and luteinizing hormone, testosterone and prolactin. A karyotype should be obtained in all adolescents with high FSH serum levels. The main causes of PA, whether or not accompanied by secondary sexual characteristics, include endocrine defects of the hypothalamus–pituitary–ovarian axis, genetic defects of the ovary, metabolic diseases, autoimmune diseases, infections, iatrogenic causes (radiotherapy, chemotherapy), environmental factors and Müllerian tract defects. PA management depends on the underlying causes. Estrogen replacement therapy at puberty has mainly been based on personal experience. PA can be due to endocrine, genetic, metabolic, anatomical and environmental disorders that may have severe implications on reproductive health later in life. In some complex cases, a multidisciplinary team best manages the adolescent, including a pediatrician endocrinologist, gynecologist, geneticist, surgeon, radiologist, and psychologist.

**Keywords:** amenorrhea; adolescents; primary ovarian insufficiency; hypogonadotropic hypogonadism; turner syndrome





## 1. Introduction

Primary amenorrhea (PA) describes the complete absence of menses by the age of 15 years. It is a devastating diagnosis that can affect the adolescent's view of her femininity, sexuality, fertility and self-image [1].

The downward secular trend in breast development onset is well-known and contrasts with the small reduction in the median age of menarche. In an epidemiological study that included 4250 school-age girls, we found that menarche occurred at a median age of 12.5 years and that 95% of girls had reached menarche by the age of 14 years [2]. Pubertal menstrual disorders, such as heavy menstrual bleeding, are commonly observed in the first 2 years after menarche. On the other hand, prolonged amenorrhea beyond the age of 14 years is not normal and needs to be evaluated.

The menstrual cycle is considered a "vital sign". Therefore, PA is an important clinical indicator that must be investigated.

A normal menstrual cycle can occur only in the presence of:

- A properly functioning hypothalamus–pituitary axis;
- Well-developed and active ovaries;
- Outflow tract without abnormalities.

Any dysfunction in any of these players can result in amenorrhea [3].

PA concerns approximately 0.1% of girls [4]. Despite this low prevalence, PA is often the presenting sign of an underlying endocrine disease, chronic disease, gene alteration, or Müllerian duct anomalies.

PA in adolescence is likely to require multi-disciplinary management (pediatrician endocrinologist, gynecologist, surgeon, psychologist, and fertility team) and a sensitive, age-appropriated approach that takes into account the adolescents' emotional maturity.

The diagnosis must be promptly confirmed. When necessary, estrogen replacement therapy should be proposed for pubertal development and psychological improvement [5].

## 2. Definition

PA can be seen as an opportunity for the early diagnosis/treatment of conditions that affect the hypothalamus–pituitary–ovarian axis [6]. It may be associated with significant morbidity and can allow for the early identification and management of health issues with potential negative consequences for adult life [7]. Indeed, the menstrual cycle should be considered as a "vital" sign.

PA is a risk factor of early and late consequences, in function of the degree of estrogenization. In adolescents with high estrogen levels, peri-pubertal hyperestrogenism constitutes a risk of endometrium hyperplasia that causes dysfunctional uterine bleeding. Later in life, it is a risk of endometrial cancer and breast cancer. Conversely, in estrogen-deficient adolescents, a reduction in bone mineral density increases the life-long risk of bone fractures [8] and of cardiovascular diseases [9]. In addition, psychological problems and even psychiatric disorders should not be overlooked.

There is no consensus on which PA type requires investigations. In our experience, PA management should be started in four conditions:

- Adolescent who did not reach menarche by the age of 14 years;
- Adolescent who did not reach menarche after more than 3 years since thelarche occurrence;
- Adolescent who did not reach menarche by the age of 13 years and without secondary sexual characteristic development;
- Adolescent who did not reach menarche by the age 14 years and with suspected eating disorder or excessive exercise, with signs of hyperandrogenism, or with failure to thrive.

## 3. Causes of Primary Amenorrhea

According to the American Society for Reproductive Medicine (ASRM) practice committee [10], PA can be associated with anatomic defects of the outflow tract, primary hypogonadism (XX, X0, XY), hypothalamic causes (dysfunction, Kallman syndrome, chronic illness), pituitary disorders (prolactinoma, infection diseases), other endocrine gland disorders (adrenal, thyroid, ovary), or multifactorial causes (polycystic ovary syndrome, PCOS). In our opinion, this classification does not reflect routine practice. We prefer to consider that PA could be related to endocrine defects (hypothalamus–pituitary–ovarian axis alterations), genetic abnormalities, previous radiotherapy or/and chemotherapy, metabolic diseases, autoimmune disorders, infections, exposure to endocrine-disrupting chemicals (EDCs), Müllerian tract defects, or unknown factors.

In our experience, in recent decades, PA was mainly explained by outflow tract anatomic defects (10%), ovarian causes (30%), pituitary causes (5%), hypothalamic causes (10%), functional causes (30%), and unknown causes (30–35%).

In the literature, PA is mainly caused by (Table 1):

1—Endocrine defects of the hypothalamus–pituitary–ovarian axis.

2—Genetic defects of the ovary.
3—Metabolic diseases.
4—Autoimmune diseases.
5—Infections.
6—Iatrogenic causes (radiotherapy, chemotherapy).
7—Müllerian tract defects.
8—Environmental factors.
9—Idiopathic

**Table 1.** The main causes of adolescent primary amenorrhea.

| **1—Endocrine Defects within the Hypothalamo–Pituitary–Ovarian Axis (HPOa)** |
| --- |
| *(1).*    *Hypogonadotropic hypogonadism* |
| *(a)*    Congenital hypogonadotropic hypogonadism |
| ▪    *Kalman syndrome* |
| ▪    *Prader–Willi syndrome* |
| ▪    *Septo-optic dysplasia* |
| *(b)*    Acquired hypogonadotropic hypogonadism |
| ▪    *Brain tumor* |
| ▪    *CNS infiltration diseases* |
| ▪    *Chemo-radiotherapy* |
| ▪    *Hyperprolactinemia* |
| *(c)*    Functional hypothalamic amenorrhea |
| *(2).*    *Hypergonadotropic hypogonadism* |
| *(a)*    XX |
| –    *Congenital* |
| ●    *Premature ovarian insufficiency* |
| –    *Acquired* |
| ●    *Chemo-radiotherapy* |
| ●    *Metabolic disorders* |
| ●    *Auto-immune disease* |
| ●    *Infections* |
| ●    *Environmental disrupting chemicals* |
| *(b)*    X0 |
| *Turner syndrome* |
| *(b)*    XY |
| –    *Gonadal dysgenesis* |
| –    *Androgen resistance syndromes* |
| **2—Müllerian defects (Normogonadotropic amenorrhea)** |

*3.1. Endocrine Defects of the Hypothalamus–Pituitary–Ovarian Axis*

When puberty does not occur at the extreme end of the normal spectrum (i.e., constitutional delay of growth and puberty), hypogonadism must be suspected. There are two types of hypogonadism: hypogonadotropic hypogonadism and hypergonadotropic hypogonadism. Hypogonadotropic hypogonadism can be transient due to an underlying medical condition, or persistent due to a gonadotropin-releasing hormone (GnRH) defect. Midline congenital defects, such as cleft lip and palate, and neural tube defects are suggestive of permanent hypogonadotropic hypogonadism. Hypergonadotropic hypogonadism is due to gonad failure.

### 3.1.1. Hypogonadotropic Hypogonadism

Hypogonadotropic hypogonadism (CHH) may be congenital and due to isolated FSH-LH deficiency or multiple gonadotropin deficiencies. It may be secondary to an acquired hypothalamic–pituitary disease. Hypothalamic–pituitary diseases include Kallmann syndrome, Prader–Willi syndrome, congenital hypopituitarism, and septo-optic dysplasia, among others.

Congenital Hypogonadotropic Hypogonadism

Hypogonadotropic hypogonadism may be due to gene alterations, and their identification facilitates a differential diagnosis and then management:

- The genes encoding kisspeptin and its receptor (*KISS1* and *KISS1R*) and neurokinin B and its receptor (*TAC3* and *TACR3*), which regulate GnRH release, should be the first to be screened in clinical settings for equivocal cases, such as delayed puberty versus idiopathic hypogonadotropic hypogonadism, because they are the main causes of GnRH pulse generator defects.
- In Kallmann syndrome, the screening of specific genes should be prioritized based on their association with clinical features: synkinesis (*KAL1*), dental agenesis (*FGF8/FGFR1*), bone anomalies (*FGF8/FGFR1*), and hearing loss (*CHD7, SOX1*). New genes have been recently identified and the list of genes involved in hypogonadotropic hypogonadism is still growing.

More than 25 different genes have been implicated in congenital hypogonadotropic hypogonadism and Kallmann syndrome, which account for ~50% of CHH cases.

Acquired Hypogonadotropic Hypogonadism

The absence of maturation of the hypothalamic–pituitary–ovarian axis may be secondary to acquired hypothalamic–pituitary diseases, such as:

- Brain tumors: craniopharyngioma, astrocytoma;
- Central nervous system infiltration diseases (e.g., histiocytosis);
- Chemo- or radiotherapy;
- Hyperprolactinemia [11].

Functional Hypothalamic Amenorrhea

The diagnosis of functional hypothalamic amenorrhea (FHA) is carried out after having excluded all anatomic and organic causes of PA [12]. According to Gordon et al., 2017, FHA is often associated with stress, weight loss, excessive exercise, and their combination [13].

The evaluation of adolescents with suspected FHA includes a detailed history focused on diet, eating disorders, exercise, athletic training, perfectionism, and ambition.

### 3.1.2. Hypergonadotropic Hypogonadism

Hypergonadotropic hypogonadism can be congenital or acquired. In function of the karyotype, it can classified in:

- XX hypergonadotropic hypogonadism;

- X0 hypergonadotropic hypogonadism;
- XY hypergonadotropic hypogonadism.

XX Hypergonadotropic Hypogonadism

(a)     Premature ovarian insufficiency

Premature ovarian insufficiency (POI) ranges from 1 in 100 to 1 in 10,000 for women aged younger than 40 years with increasing prevalence at each decade of life and is specifically uncommon in adolescents. POI is characterized by severe estrogen deficiency due to a loss of ovarian function, which can be congenital or acquired [14]. The prevalence of known gene alterations that may be linked to PA is estimated at ~20%. To date, 18 POI-causing genes have been identified: *BMP15, DMC1, EIF2S2, FIGLA, FOXL2, FSHR, GDF9, GPR3, HFM1, LHX8, MSH5, NOBOX, NR5A1, PGRMC1, STAG3, XPNPEP2, BHLB*, and *FSHB*.

In a recent analysis of these genes by next-generation sequencing in patients with PA, Eskenazi et al. showed that 25% of these adolescents presented at least one variant, and 18% presented a variant of unknown significance. In this study, *NOBOX* variants were the most frequently detected (19% of all patients) [15]. Ghosh et al. confirmed that chromosomal anomalies are one of the major causes of amenorrhea in India [16]. *BMP15* mutations and *FMR1* pre-mutation (fragile X syndrome) have been detected in ~10% of adolescents with PA [15].

(b)    Metabolic disorders

- Classic galactosemia affects around 1/25,000 of new-born girls and is due to mutations in the *GALT* gene that decrease/abolish galactose-1-phosphate uridy-lyltranserase activity, leading to the toxic accumulation of galactose in the ovaries and in the whole body. Several mechanisms have been postulated, including the direct toxicity of galactose to oocytes and follicles, leading to accelerated atresia of the ovarian pool [17].
- Thalassemia and sickle cell disease are the most prevalent inherited hemoglobin disorders (recessive pattern). Transfusion-related iron overload may lead to gonad dysfunction, the absence of puberty development, and PA [18].
- Other metabolic disorders also may be associated with PA [19], such as congenital adrenal hyperplasia (due to 17-hydroxylase enzyme deficiency) and aromatase deficiency. In some girls, PA may be associated with type 1 diabetes, low BMI and abnormal pulsatile GnRH secretion.
- Obesity: adolescents with severe obesity usually have elevated levels of plasma androgens. The normalization of plasma androgens upon weight loss leads to the resumption of ovulation, suggesting that overweight-related hyperandrogenism is the cause of amenorrhea in adolescent girls with obesity [20].
- PCOS: It is not rare that PA may lead to the detection of PCOS, especially when severe insulin resistance is present in the peri-pubertal period in adolescents with central obesity, early hyperinsulinism and insulin resistance [21,22]. Many PCOS features appear during early adolescence, such as oligomenorrhea, heavy menstrual bleeding, and signs of hyperandrogenism. PA is an uncommon manifestation of PCOS (1.4 to 14% of girls present PA as an initial feature of PCOS) [21]. Early hyperinsulinism, insulin resistance, and central obesity are observed in girls with severe PCOS [22].

(c)     Autoimmune diseases

Autoimmune diseases (diabetes mellitus, hypothyroidism, Hashimoto thyroiditis, Grave disease) are common disorders associated with PA [23].

Evidence for an autoimmune mechanism involved in POI is based on the description of lymphocytic oophoritis, associated with autoantibodies against ovarian antigens [24]. PA is also associated with other autoimmune disorders of the adrenal gland, thyroid, and pancreas. Autoimmune polyglandular syndrome type 1, caused by *AIRE* gene mutations, is

also associated with PA. Autoimmune disorders can also be associated with non-endocrine diseases, such as candidiasis, vitiligo, systemic lupus, rheumatoid arthritis.

(d)   Infections

Young women living with HIV are 70% more likely to experience amenorrhea.

The overall prevalence of amenorrhea among women with HIV is ~5%. In a recent meta-analysis, King et al. showed a significant association between HIV and amenorrhea (OR = 1.68) [25]. It is unclear whether amenorrhea might be a complication of HIV infection or of other risk factors, such as low BMI, wasting (collapse of GnRH secretion), opiate and anti-psychotic use (anovulation), immunosuppression and chemotherapy. According to Cejtin et al., amenorrhea is reversible in 37% of women with HIV reporting this problem [26].

(e)   Iatrogenic causes (radiotherapy, chemotherapy)

Modern pediatric cancer management is becoming highly effective, and therefore, it is crucial to monitor the endocrine and gynecological consequences in adolescents who underwent such treatments [27,28]. It is thought that in this population, the prevalence of ovarian failure is ~10%, although Jablonska et al. found that 31.6% of cancer survivors experienced amenorrhea [29], and Chemaitilly et al. reported ovarian insufficiency rates from 2.1 to 92.2% [30]. In the ovarian follicles, both oocytes and granulosa cells are vulnerable to chemotherapy effects. In addition, damage to blood vessels and focal fibrosis of the ovarian cortex are involved in chemotherapy-induced ovarian damage [31].

Besides chemotherapy with alkylating agents, abdominal/pelvic radiation must be considered a risk factor. In adolescents who received total body irradiation before stem cell transplant, AMH, which is considered a good marker of the ovarian reserve [32], was undetectable in serum [33]. Oocytes are very sensitive to radiotherapy. The detrimental effects are influenced by the irradiation field, dose and fractionation schedules. AMH levels are lower in children who received radiotherapy to the abdomen, pelvis and total body for cancer management [34].

Lastly, in children, endocrine dysfunction following brain injury may include PA [35].

(f)   Environmental factors (lifestyle, EDCs)

Many EDCs target the ovary, with effects particularly on folliculogenesis and steroidogenesis during fetal life [36]. Several experimental data demonstrated EDC harmful effects on follicle growth by promoting atresia [37]. Moreover, some EDCs (i.e., pesticides, phthalates, bisphenol-A, and dioxin) alter ovarian steroidogenesis. Severe acute and chronic prenatal exposure to EDCs may impair ovarian function later in life and disturb puberty [38].

(g)   Idiopathic

In the past, not informing young patients about the diagnosis and treatment was the standard practice. It is now established practice to gradually disclose the diagnosis and the underlying cause to adolescents, in function of their level of understanding and knowledge. This leads to better adherence to the treatment and allows for the screening of other family members, if relevant.

X0 Hypergonadotropic Hypogonadism: Turner Syndrome

Turner syndrome is the most common example of hypergonadotropic hypogonadism and explains up to 30% of all PA. It is associated with X chromosome numerical or structural alterations. Its prevalence is ~50/100,000 girls in Caucasian populations. Although 20% of patients with Turner syndrome begin puberty spontaneously, only very few progress to menarche. The absence of puberty and PA are very frequent clinical manifestations of Turner syndrome [39]. Ovarian dysgenesis and early follicular apoptosis are key features of this syndrome, resulting in POI with estrogen deficiency in the peri-pubertal period [40]. AMH plasma concentration is a useful marker of ovarian function.

XY Hypergonadotropic Hypogonadism: Disorders of Sex Development (DSD)

The term 46, XY DSD is used to describe 46, XY adolescents with undermasculinization, leading in some cases to a female phenotype. In Denmark, the prevalence of 46, XY females was estimated to be 6.4 per 100,000 live born females [41]. Testosterone, AMH, FSH and LH serum concentration and the presence of Müllerian derivatives on pelvic ultrasound are used to differentiate gonad dysgenesis (associated with the insufficient gonadal secretion of testosterone and AMH) from androgen production defects or androgen resistance (Figure 1).

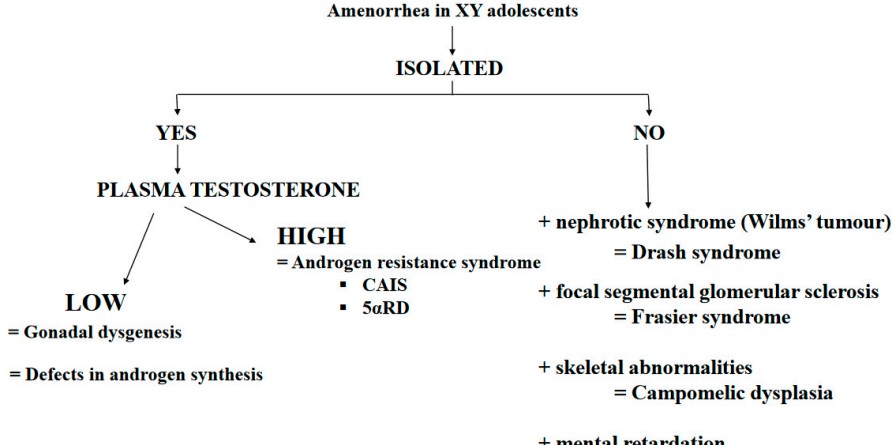

**Figure 1.** Main causes of amenorrhea in XY adolescents. Abbreviation: CAIS = complete androgen insensitivity syndrome; 5αRD = 5α-reductase deficiency.

(a)  Gonad dysgenesis:

Gonad dysgenesis is a genetic defect in testis determination characterized by variable alterations of Leydig and Sertoli cell function. This disorder can be secondary to mutations in any of the several genes implicated in the primitive gonad differentiation to testes.

*SRY* gene alterations lead to 46, XY sex reversal with a female phenotype. The diagnosis of Swyer syndrome is made at puberty in girls with PA [42].

In some cases, gonad dysgenesis has been linked to mutations of *SF1*, a gene involved in the development of male gonads and adrenal glands.

In some patients, gonad dysgenesis is associated with kidney dysfunction. In these patients, the diagnosis of Drash syndrome (i.e., Wilms' tumor and kidney insufficiency) or Frasier syndrome (i.e., proteinuria secondary to focal glomerular sclerosis) may be made. Both syndromes are due to disease-specific *WT1* gene abnormalities. Indeed, heterozygous mutations in the open reading frame have been associated with Drash syndrome, while intron mutations leading to splicing abnormalities have been found in patients with Frasier syndrome.

Several *SOX9* mutations have been identified in patients with severe skeletal malformations (e.g., campomelic dysplasia) associated with sex reversal and gonad dysgenesis.

Homozygous or composite heterozygous mutations of the desert hedgehog (*DHH*) gene, which is implicated in testis differentiation and perineal development, have been identified. These patients present a female phenotype, sometimes with neuropathy.

(b)  Testosterone Production Defects

Defects in testosterone production are rare and are characterized by severe external genital undervirilization. Conversely, no Müllerian derivatives are present because AMH is normally secreted by Sertoli cells. These defects are due to an enzymatic defect in testosterone biosynthesis or are secondary to alterations in the gene encoding the LH receptor.

Leydig cell agenesis (or hypoplasia) is a rare form of 46, XY DSD. The typical presentation is PA, no breast development at puberty, and low testosterone at baseline and after human chorionic gonadotropin (hCG) stimulation. This condition is determined by a homozygous or double heterozygous inactivating mutation of the LH receptor gene.

Defects in 3-β-hydroxysteroid dehydrogenase are associated with variable, but insufficient virilization in 46, XY boys, ranging from a female phenotype to minor DSD forms.

Similarly, 17-α-hydroxylase defects lead to a variable phenotype. In some patients, the diagnosis is made only at puberty due to PA. An excess of 11-deoxycorticosterone due to *CYP17* gene alterations (recessive transmission) causes hypertension during puberty.

Defects of 17-β-hydroxysteroid reductase are rare, but cause testicular block and deficits in testosterone production, leading frequently to a female phenotype.

(c)    Androgen-resistance disorders

Androgen-resistance disorders are characterized by normal/high testosterone and AMH production. These disorders are caused by an androgen receptor (AR) defect or 5α-reductase deficiency.

- Complete androgen insensitivity syndrome (CAIS)

CAIS is often diagnosed during puberty in girls with PA, normal breast development, and sparse axillary and pubic hair [43]. The endocrine work-up shows high plasma testosterone and LH concentrations. The diagnosis is confirmed by the identification of an AR gene mutation.

- 5α-reductase deficiency

The phenotype is typically female, but all degrees of undervirilization can be observed [44,45]. The diagnosis is usually made at puberty because of PA, the absence of breast development, and striking virilization (hirsutism, clitoral hypertrophy, significant muscle development, and behavior masculinization). The diagnosis is confirmed by identifying a mutation in the gene encoding 5α-reductase 2.

*3.2. Normogonadotropin Ovulatory Amenorrhea: Congenital Müllerian Defects*

Müllerian anomalies concern 4 to 5% of women [46]. Several classifications have been proposed, but without a consensus. The most used and accepted was proposed by the ASRM and classifies these anomalies in 12 classes [47].

Congenital malformations of the female genital organs include the absence of a uterus and vagina, and some obstructive abnormalities of the reproductive tract [48]. Müllerian aplasia or hypoplasia, also known as Mayer–Rokitanski–Kuster–Hauster syndrome, may be isolated or associated with other congenital malformations [49].

Transverse vaginal septum is caused by the persistence of the vaginal plate after it meets the Müllerian tract. Examination reveals a shortened, blind vaginal pouch.

An imperforated hymen usually presents as a bluish bulging mass due to hematocolpos at the vagina entrance.

## 4. Primary Amenorrhea Evaluation

In the presence of an adolescent girl with PA, the diagnosis can be guided by her history, physical examination, imaging studies, hormone evaluation, and karyotyping [1,50]. First and foremost, it is imperative to rule out pregnancy.

(a)    PA evaluation begins by collecting the patient's medical history (general health and lifestyles), particularly to identify chronic diseases and exposure to radiation or chemotherapy during infancy. It is important to obtain information about the history of galactorrhea, headache, and cyclical abdominal pain.

(b)    A physical examination must include height and weight, and a body mass index (BMI) calculation. The Tanner stage of breast development is a good marker of the degree of estrogenization [19]. Features suggestive of Turner syndrome must be recorded. A scrupulous examination of the external genitalia and cervix should be conducted [5].

(c)    Imaging studies routinely include pelvic ultrasonography to confirm the presence of ovaries and the uterus.

(d)    The initial hormone evaluation is limited to the serum-follicle-stimulating hormone (FSH) and luteinizing hormone (LH), testosterone and prolactin [10]. Pregnancy must

be ruled out because adolescents may ovulate before the first period. A karyotype should be obtained in all adolescents with high FSH serum levels.

At the end of this evaluation, PA causes should be discussed (Figure 2):

- In function of the initial examination: PA with or without breast development, with or without evidence of androgen excess, with or without galactorrhea, with or without weight loss, with or without growth failure.
- In function of the FSH levels: hypergonadotropic hypogonadism (elevated FSH), hypogonadotropic hypogonadism (low FSH) or eugonadism (normal FSH).
- In function of the karyotype: XX, XO or XY [51].

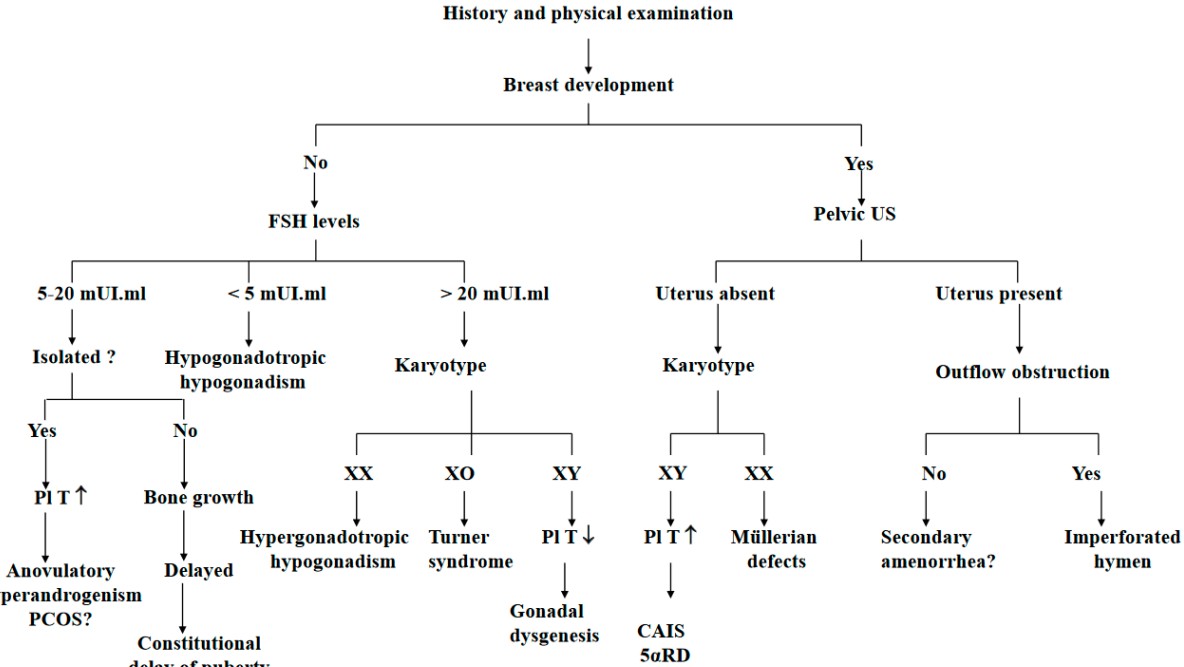

**Figure 2.** Algorithm for the evaluation of primary amenorrhea. Abbreviation: FSH = follicle-stimulating hormone; US = ultrasonography; Pl = plasma; T = testosterone concentration; MRKH = Mayer–Rokitanski–Kuster–Hauster syndrome; CAIS = complete androgen insensitivity syndrome; 5αR = 5α-reductase.

## 5. Management

PA management depends on the underlying causes and the patient's health status, psychological concerns, and goals [52].

The overall goal of estrogen replacement therapy in girls with hypogonadism is to start the development and maturation of secondary sexual characteristics and uterine growth, ensure normal growth velocity and optimal bone mass acquisition and to reduce the psychological consequences of the lack of this hormone [53].

Estrogen replacement therapy at puberty has mainly been based on personal experience because no guideline has been published on the estrogen type, treatment route, dose and time [54].

Some authors proposed a short-term test with low estrogen doses (2–6 mg/day) for 6–12 months. Although there is no consensus about this procedure, we follow this advice. There are various treatment protocols: estrogen therapy is routinely initiated at the age of 12–13 years, at a low dose (approximately 1/10 of the adult dose), and gradually increased over the next 2–4 years. Transdermal administration (patch or gel) seems more physiological.

In adolescents who did not develop breasts; first, a very low dose of estrogen should be used to mimic the gradual onset of puberty maturation. The typical regimen consists of

an estrogen equivalent of 6 µg/daily for 6 months, with a dose increase every 6 months until the second year. When vaginal bleeding begins, cyclic progesterone therapy is added (12–14 days per month).

Regardless of the form, administration route and dose, appropriate estrogen therapy is introduced also to prepare patients for assisted reproductive procedures. In adult life, the most frequent therapeutic infertility approach is embryo transfer from donated oocytes. Moreover, the reconstitution of complete oogenesis from induced pluripotent stem cells could be envisaged in the future [55].

In girls with an imperforated hymen, a cruciate incision is made to open the vaginal orifice. In the presence of transverse septum, surgical removal is required. Cervix hypoplasia or absence is more difficult to treat. If the vagina is short, progressive dilatation is required.

## 6. Conclusions

Primary amenorrhea can be due to endocrine, genetic, metabolic, anatomical and environmental disorders that may have severe implications on reproductive health later in life.

In some complex cases, a multidisciplinary team best manages the adolescent, including a pediatrician endocrinologist, gynecologist, geneticist, surgeon, radiologist, and psychologist.

Delay in the evaluation (and treatment) of adolescent amenorrhea in some cases may contribute to reduced bone density and other long-term adverse health consequences.

XY adolescents with a female phenotype are non-exceptional conditions and should be managed in reference centers [56].

Next-generation sequencing for gene screening should be proposed to all adolescents with amenorrhea and idiopathic POI.

Exposure to EDCs during fetal life, childhood and adolescence is a potential risk factor of adolescent amenorrhea.

**Author Contributions:** L.G., F.P., N.K. and C.S.: managed patients according to their personal experience; L.G.: drafted the manuscript; F.P., N.K. and C.S.: revised the manuscript critically for intellectual content. All authors have read and agreed to the published version of the manuscript.

**Funding:** The authors received no specific funding for this work.

**Data Availability Statement:** Data regarding any of the subjects in the study have not been previously published.

**Acknowledgments:** We thank Elisabetta Andermarcher for the editing of this work.

**Conflicts of Interest:** The authors have no conflict of interest to declare concerning this paper that could be perceived as prejudicing the impartiality of the review reported.

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
