# Peer review of "Primary Amenorrhea in Adolescents: Approach to Diagnosis and Management"

_endocrines, doi:10.3390/endocrines4030038_

Round 1
Reviewer 1 Report
Summary
This review article analysis the issue of primary amenorrhea (PA) in adolescents. PA may be an important clinical indicator of underlying disorders (endocrine and metabolic diseases, genetic abnormalities, iatrogenic and environmental causes), that requires evaluation. The authors aimed to propose a diagnostic algorithm, present causes and management of this disorder.
General concept comments
The layout of the article is correct: the definition, general causes and the diagnostic algorithm of primary amenorrhea proposed by the authors are presented. Subsequently, diseases resulting in PA are discussed. The review is quite brief, it does not contain details of the presented diseases, but rather signals their presence, what can be both advantage and disadvantage of paper.
The PubMed database has 3,648 results on the topic of primary amenorrhea. The diagnostic algorithm proposed by the authors differs only slightly from the schemes presented in other studies, for example:
-
Seppä S, Kuiri-Hänninen T, Holopainen E, Voutilainen R. MANAGEMENT OF ENDOCRINE DISEASE: Diagnosis and management of primary amenorrhea and female delayed puberty. Eur J Endocrinol. 2021 May 4;184(6):R225-R242. doi: 10.1530/EJE-20-1487. PMID: 33687345.
-
Klein DA, Paradise SL, Reeder RM. Amenorrhea: A Systematic Approach to Diagnosis and Management. Am Fam Physician. 2019 Jul 1;100(1):39-48. PMID: 31259490.
-
Master-Hunter T, Heiman DL. Amenorrhea: evaluation and treatment. Am Fam Physician. 2006 Apr 15;73(8):1374-82. PMID: 16669559.
What is more, the studies available in the literature present the discussed topic in a more detailed way.
The advantage of this paper is the presence of a concise summary of treatment methods as a separate paragraph.
Specific comments
-
Many relevant citations are omitted:
-
Line 71: …. Cardiovascular diseases.
-
Lines 113-174,
-
Lines 213-220.
-
-
Information given in line 117 is not specific: „pituitary disorders (prolactinoma, disease)”- what kind of disease?
-
Causes of PA presented in lines 124-126 should be displayed in a descending order and relevant citations should be given.
-
Data from lines 128-136 is a repetition of the information on lines 119-123.
-
The content of table 1 is inconsistent with the data in the text: Muller tract defects are presented in line 134 as the seventh reason of PA and in Table 1 are the second cause. What is more, table title is “Main causes of adolescent primary amenorrhea” and only two are presented.
-
Presented definition of premature ovarian insufficiency seem to be inappropriate. In line 191- “ovarian dysgenesis” should be changed into “loss of ovarian function”.
-
Pervelance of POI is quoted wrongly. In cited article: “The prevalence of POI ranges from 1 in 100 to 1 in 10,000 for women aged younger than 40 years with increasing prevalence at each decade of life and is specifically uncommon in adolescents” – what suggests even lower pervalance of this disorder in discussed age group.
-
Lactational amenorhhea mentioned in lines 229-234 is not a metabolic disorder and is more suitable for secondary amenorrhea diagnostic algorithm.
-
Line 241-242: Mutation of AIRE gene causes only type 1 of autoimmune polyglandular syndromes. Other types are not monogenic.
Author Response
Specific comments
- Many relevant citations are omitted:
- Line 71: …. Cardiovascular diseases = OK, we have included this reference: “The long-term cardiovascular risks associated with amenorrhea” by Simoncini et al.
- Lines 113-174, = OK, we have included this reference: “Current evaluation of amenorrhea” by Practice committee and “Hyperprolactinemia and “Pituitary Causes of Amenorrhea” by Fazeli and Nachtigall.
- Lines 213-220. = OK, we have included this reference: “Evaluation and management of amenorrhea related to congenital sex hormonal disorders” by Yoon and Cheon and “The effects of obesity on the menstrual cycle” by Itriyeva.
- Information given in line 117 is not specific: „pituitary disorders (prolactinoma, disease)”- what kind of disease? OK, we have replaced “disease” by “infection diseases”.
- Causes of PA presented in lines 124-126 should be displayed in a descending order and relevant citations should be given. No, this sentence is related to our own clinical experience.
- Data from lines 128-136 is a repetition of the information on lines 119-123. No, conversely to the previous sentence, this part is related to the literature data.
- The content of table 1 is inconsistent with the data in the text: Muller tract defects are presented in line 134 as the seventh reason of PA and in Table 1 are the second cause. What is more, table title is “Main causes of adolescent primary amenorrhea” and only two are presented. No: main causes of adolescent PA should be related to endocrine defects with in the HPO axis and Mullerian defects. There is no controversy in this condition.
- Presented definition of premature ovarian insufficiency seem to be inappropriate. In line 191- “ovarian dysgenesis” should be changed into “loss of ovarian function”. OK, we have followed the reviewer’s advice.
- Pervalance of POI is quoted wrongly. In cited article: “The prevalence of POI ranges from 1 in 100 to 1 in 10,000 for women aged younger than 40 years with increasing prevalence at each decade of life and is specifically uncommon in adolescents” – what suggests even lower pervalance of this disorder in discussed age group. Thanks for this good comment.
- Lactational amenorhhea mentioned in lines 229-234 is not a metabolic disorder and is more suitable for secondary amenorrhea diagnostic algorithm. OK, we have dropt this sentence.
- Line 241-242: Mutation of AIRE gene causes only type 1 of autoimmune polyglandular syndromes. Other types are not monogenic. OK, we have modified the sentence accordingly.
Reviewer 2 Report
Manuscript Title: Primary Amenorrhea in Adolescents: Approach to Diagnosis and Management
Dear Editor,
Although primary amenorrhea affects less than 1% of adolescent girls it may have inevitable physical and psychological consequences. This review is written by a professional basic scientific group with experience in the field of early sex development. It would have been better if the experience of clinicians was used, especially in the management section. There are many similar articles concerning this issue in the literature. They must have a compelling reason to write this review and have something new. Unfortunately, I could not find such things in the manuscript.
General
1- I believe the Primary amenorrhea description has an age cut-off. Please include the age in the abstract and the main text.
2- The English language within your manuscript needs improvement and it is better to have your manuscript reviewed by someone fluent in English.
Abstract
1- In the classification, it must be mentioned whether or not it is accompanied by secondary sexual characteristics.
Body
1- Each sentence in the introduction and discussion part should have a reference (s). For example, “PA concerns approximately 1-5% of girls” needs a reference.
2- I thank it is better to replace section 2 (evaluation) with section 3 (etiology).
Tables
1- Please mention the reference of the Table 1. What do you mean of acquired? Chemical agents or radiotherapy are missing in the table.
Figures
1- All abbreviations in the figures should be described, for example, US or PI.
Best regards,

Author Response
General
1- I believe the Primary amenorrhea description has an age cut-off. Please include the age in the abstract and the main text. OK, we have done.
2- The English language within your manuscript needs improvement and it is better to have your manuscript reviewed by someone fluent in English.
Abstract
1- In the classification, it must be mentioned whether or not it is accompanied by secondary sexual characteristics. OK, we have done.
Body
1- Each sentence in the introduction and discussion part should have a reference (s). For example, “PA concerns approximately 1-5% of girls” needs a reference. OK, we have added some references.
2- I thank it is better to replace section 2 (evaluation) with section 3 (etiology). Following reviewer’s comment, we have replaced section 3 with section 2.
Tables
1- Please mention the reference of the Table 1. What do you mean of acquired? Chemical agents or radiotherapy are missing in the table. Sorry, there is no reference, since table 1 is based on our clinical experience. OK, we have added « •Chemo-radiotherapy » amons the causes of acquired hyper hypo.
Figures
1- All abbreviations in the figures should be described, for example, US or PI. OK, we have done.
Round 2
Reviewer 2 Report
Thank you the comments are answered properly.